# Detection and Characterization of Electrogenic Bacteria from Soils

**DOI:** 10.3390/biotech12040065

**Published:** 2023-12-02

**Authors:** Ana Rumora, Liliana Hopkins, Kayla Yim, Melissa F. Baykus, Luisa Martinez, Luis Jimenez

**Affiliations:** Biology and Horticulture Department, Bergen Community College, 400 Paramus Road, Paramus, NJ 07652, USA; arumora@me.bergen.edu (A.R.); kyim142129@me.bergen.edu (K.Y.); lmartinez144129@me.bergen.edu (L.M.)

**Keywords:** soil microbial fuel cells, electrogenic bacteria, Bacillota, Pseudomonadota

## Abstract

Soil microbial fuel cells (SMFCs) are bioelectrical devices powered by the oxidation of organic and inorganic compounds due to microbial activity. Seven soils were randomly selected from Bergen Community College or areas nearby, located in the state of New Jersey, USA, were used to screen for the presence of electrogenic bacteria. SMFCs were incubated at 35–37 °C. Electricity generation and electrogenic bacteria were determined using an application developed for cellular phones. Of the seven samples, five generated electricity and enriched electrogenic bacteria. Average electrical output for the seven SMFCs was 155 microwatts with the start-up time ranging from 1 to 11 days. The highest output and electrogenic bacterial numbers were found with SMFC-B1 with 143 microwatts and 2.99 × 10^9^ electrogenic bacteria after 15 days. Optimal electrical output and electrogenic bacterial numbers ranged from 1 to 21 days. Microbial DNA was extracted from the top and bottom of the anode of SMFC-B1 using the ZR Soil Microbe DNA MiniPrep Protocol followed by PCR amplification of 16S rRNA V3-V4 region. Next-generation sequencing of 16S rRNA genes generated an average of 58 k sequences. BLAST analysis of the anode bacterial community in SMFC-B1 demonstrated that the predominant bacterial phylum was Bacillota of the class Clostridia (50%). However, bacteria belonging to the phylum Pseudomonadota (15%) such as *Magnetospirillum* sp. and *Methylocaldum gracile* were also part of the predominant electrogenic bacterial community in the anode. Unidentified uncultured bacteria accounted for 35% of the predominant bacterial community. Bioelectrical devices such as MFCs provide sustainable and clean alternatives to future applications for electricity generation, waste treatment, and biosensors.

## 1. Introduction

Microbial activity is the driving force behind the biochemical cycling of organic and inorganic compounds in the biosphere. Processes such as carbon and nitrogen cycling are dependent on the activity of bacteria, archaea, fungi, and protozoa [1]. However, the great majority of microorganisms are not grown in laboratory cultures [2]. Microorganisms in soil are highly diverse and play an important role in the decomposition of organic matter to support plant growth, soil structure, and fertility. The composition of the microbial community in soils can be affected by biotic and abiotic factors. For instance, temperature, pH, and moisture are among the abiotic factors that control microbial activity in soils. Competition for resources, predation, and parasitism are some biotic factors that can also affect the diversity of microbes. Of all these factors, pH is the best predictor of bacterial diversity in soil. Decomposition of organic compounds is based upon the oxidation of different substances such as carbohydrates, fats, and proteins to support microbial metabolism in order to sustain growth and survival in a very dynamic and competitive matrix such as soil. Microbial decomposition of organic matter under aerobic conditions results in the complete mineralization of organic materials to carbon dioxide and water. However, when anaerobic conditions are present, microbial activity is driven by processes such as anaerobic respiration, fermentation, and methanogenesis.

Because of the energy crisis around the world due to the environmental changes caused by fossil fuel contamination and global warming, renewable energy sources are currently being explored to provide a safer alternative to develop a more sustainable society [3]. Microbial fuel cells (MFCs) are bioelectrical devices driven by the oxidation of organic matter by microorganisms. MFCs have been shown to harness the natural metabolism of microbes in soils to produce electrical power [3]. Microbial oxidation of organic compounds and the reduction of electron acceptors such as iron and nitrate in the anode are the only source of electron generation in the MFC system. Electron transfer by microbial cells to the anode electrodes can be achieved directly by transferring the electrons via bacterial cell membrane cytochromes, pili, microbial nanowires, and protein complexes. However, certain microbes transfer electrons indirectly using environmental or self-produced extracellular electron mediators [4]. Microbes from soils and sediments have been shown to generate electricity using different MFC formats [5,6,7,8]. Aerobic and anaerobic soil microbial fuel cells (SMFCs) have been constructed with single and multiple chambers, optimizing electricity generation and microbial activity. For instance, different types of graphite felt electrodes have been tested as the anode in an air-cathode and membrane-less SMFC [9]. Researchers combined a composite anode material with a conductive polymer and a transition metal oxide to enhance biofilm formation and electrogenesis. Higher biofilm formation and electrogenesis was observed with a combination of graphite felt and cobalt oxide and polyaniline as the anode. Electron microscopy analysis demonstrated the formation of dense microbial growth on the anodes with a wide variety of shapes and sizes of filamentous microorganisms.

In another study, the effect of different cathode materials was ascertained to enhance electrogenesis [10]. Graphite felt was found to be the best electrode material for optimal electrogenesis with higher voltage, catalytic activity, and a significant enrichment of bacteria belonging to the genus *Hydrogenophaga*. These bacteria can enhance cathodic activity by enhancing electron uptake through hydrogen evolution or nitrate reduction. Because the organic compounds used to generate electrical output in SMFCs eventually decreases leading to the significant reduction of electrogenic activity, a hybrid MFC with plants and soils has been shown to sustain long-term activity through the continuous supply of organic substances from plant photosynthesis [11]. The physical separation of the plant in a different compartment from the anode allows the optimization of electrogenesis by avoiding the diffusion of oxygen from the roots into the anode but allowing the continuous supply of the plant leachate. Another possible solution to the depletion of organic matter during operational time is to add an external carbon source such as compost, cellulose, or glucose to replenish the organic materials lost during microbial oxidation [12]. In the SMFCs, two of the most important parameters affecting electricity generation are the types of microbes and soil chemical composition [7,13,14,15]. Soil types and microbial dynamics can influence SMFC performance. The interaction between the different bacterial species affects the function, activity, and stability of the community, providing optimal metabolic capabilities leading to the production of electricity. Furthermore, the availability of certain types of organic compounds such as polyphenols may limit the production of electrical power by bacterial communities. Forest and agricultural soils were shown to be completely different when it came to developing and sustaining an electrogenic bacterial community capable of significant electrical production [6]. Agricultural SMFCs showed 17 times more electricity and 10 times higher respiration rates than forest SMFCs. High clay content has been shown to make soil less porous and permeable to oxygen diffusion from the cathode to the anode, optimizing the anaerobic conditions for the microbial communities in the anode [13]. SMFCs with high clay content demonstrated higher electrogenic activity and longer and more sustainable operational times [13]. Furthermore, SMFCs with high organic matter exhibited faster start-up times and higher electrogenic activity. Start-up operational times are defined as the enrichment process when bacteria adapt to anaerobic conditions and develop an optimal biomass on the anode surface resulting in electrical output by the oxidation of natural substances.

A wide variety of bacteria have been shown to be associated with electrodes in MFCs. Bacterial belonging to the phyla Pseudomonadota, Bacillota, Actinomycetota, and Bacteroidota were shown to be electrogenic in MFCs [3]. Specific bacterial species seemed to be more electrogenic than others. For instance, *Geobacter sulfurreducens* is a very efficient electrogenic bacteria [5]. Pure and mixed cultures of *G. sulfurreducens* have been used to generate electricity in SMFCs. *G. sulforreducens* has been shown to reduce Fe^+3^ using a variety of organic compounds as electron donors. As of now, *G. sulfurreducens* is the most electrogenic bacterium with the longest viability and activity in MFCs [16]. Pure cultures of *G. sulforreducens* were capable of generating electricity as efficiently as mixed bacterial cultures. Electron transfer is prevalently performed through the outer membrane cytochromes and iron-containing proteins that connect the bacteria directly to the anode. It can be also mediated by pili. However, not having a pilus significantly reduced electrical production. However, the presence of pili is needed for optimal electrical production. Other electrogenic bacteria with high electrical production are members of the genus *Shewanella* [3]. *Shewanella* can transfer the electrons directly to the anode through the outer membrane cytochromes but it also uses pili. Genera in the phylum Bacillota such as *Bacillus* and *Clostridium* have been shown to be important members of the electrogenic bacterial community in SMFCs [3,6]. *Clostridium* can also transfer electrons directly to the anode without having any electron transfer mediator.

Previous studies in our laboratory demonstrated a sustainable electrical production lasting a maximum of 23 days with a power output of 73 microwatts [15]. The maximum power output by SMFCs was reported to be 80 microwatts but it lasted only 12 days. 16S rRNA analysis showed that the most abundant bacteria in the anodes were members of Pseudomonadota, Bacillota, Actinomycetota, Chloroflexota, and Planctomycetota. SMFCs lacking large numbers of bacteria belonging to Bacillota did not generate electricity. However, only six soil samples from different locations were used to build the SMFCs [15]. There must be other sites around the Bergen Community College campus (BCC) or nearby locations with soils that might provide a greater potential to develop a better SMFC with longer operational time and higher electrical output. The major objective of this study was to analyze different soils around the BCC campus or nearby locations to determine the microbial communities’ potential to generate electricity and to identify electrogenic bacteria directly through 16S rRNA analysis.

## 2. Materials and Methods

### 2.1. Soil Sampling

Seven surface soils were collected from different locations at BCC and 2 from Saddle River County Park located in the city of Paramus, New Jersey, USA. Samples were aseptically taken as previously described [15]. Each type of soil was immediately used to make mud suspensions as described below.

### 2.2. Microbial Fuel Cell Assembly

Mud suspensions in deionized water were constructed using soils. The mud suspensions were placed into the MudWatt cells (Figure 1) [15]. The electrodes were constructed from a circular carbon cloth. The cylindrical MFC was made of a plastic material with a plastic lid. For each SMFC, about 1 cm of soil was placed at the bottom of the plastic container before installing the anode; additional soil was added on top of the anode until the SMFC was 90% full. The cathode was placed on top of the soil. The hacker board was placed on the indentation of the lid. The board has a microchip that takes the power generated by the MFC and converts the voltage to 2.4 Volts in short bursts, which powers the light-emitting diode (LED). The anode and cathode were connected to the hacker board and the lid was attached to seal the container. Finally, the LED and capacitor were connected to the hacker board and the SMFCs were incubated at 35–37 °C.

### 2.3. Electricity and Electrogenic Bacteria Measurements

The electrical power output and numbers of electrogenic bacteria were measured using an Application (App) downloaded onto an iPhone 14. The App was developed by Keego Technologies (http://www.mudwatt.com) (accessed on 10 November 2023) and was freely available from the Apple App Store.

### 2.4. DNA Extraction and PCR Analysis of Bacterial 16S rRNA Genes in SMFC Samples

Microbial DNA from the biofilm grown on and under the anode in SMFC-B1 was extracted using the ZR Soil Microbe DNA MiniPrep Protocol (Zymo Research, Irvine, CA, USA). Approximately 250 milligrams of the biofilm grown on the anode was aseptically added to BashingBead^TM^ Lysis Buffer. Samples were mixed for 5 min at maximum speed. After mixing, centrifugation was performed at 10,000× *g* for 3 min.

After centrifugation, 400 microliters of supernatant were added to a Zymo-Spin^TM^ III-F Filter in a collection tube followed by an additional centrifugation step at 10,000× *g* for 1 min. The filtrate was treated with 800 microliters of Genomic Lysis Buffer and 400 microliters of 95% ethanol. After mixing, samples were transferred to a Zymo-Spin IIC Column placed in a collection tube followed by centrifugation at 10,000× *g* for 1 min. Sample flow was discarded with the extracted DNA bound to the filters. A Pre-Wash Buffer was added to the filter followed by centrifugation at 10,000× *g* for 1 min. The flow was discarded and 500 microliters of g-DNA Wash Buffer were added followed by centrifugation at 10,000× *g* for 1 min.

The Zymo-Spin IIC Column with the DNA were transferred to a new collection tube and 100 microliters of Elution Buffer were added directly to the column matrix. Samples were centrifuged at 10,000× *g* for 30 s to elute the DNA from columns. A Zymo-Spin^TM^ III-HRC Filter was treated with 600 microliters of Prep Solution followed by centrifugation at 8000× *g* for 3 min. To further purify the eluted DNA, samples were added to the Zymo-Spin^TM^ III-HRC Filter in a clean collection tube. Purified DNA was suitable for PCR analysis and cloning.

DNA concentration was determined by using the Qubit^®^ dsDNA HS assay as previously described by Jimenez et al. [18]. PCR amplification of extracted DNA was performed using primers 341f (CCTACGGGNGGCWGCAG) and 785r (GACTACHVGGGTATCTAATCC), which amplified the V3-V4 fragment of the 16S rRNA gene with a size of approximately 465 base pairs (bp). Reaction conditions were as follows: 95 °C for 5 min, followed by 25 cycles consisting of denaturation at 95 °C for 40 s, annealing at 55 °C for 2 min, and extension at 72 °C for 1 min. After the 25 cycles were completed, a final extension step at 72 °C for 7 min was added to the reaction [19]. Ready-To-Go (RTG) PCR beads (GE Healthcare, Buckinghamshire, UK) were used for each PCR reaction volume as previously described [15]. Reaction mixtures were added to the T100TM thermal cycler (Bio-Rad Laboratories, Hercules, CA, USA) or Mastercycler thermal cycler (Eppendorf Scientific, Westbury, NY, USA). After PCR amplification, amplicon detection was analyzed by gel electrophoresis using the FlashGel system (Lonza Inc., Rockland, ME, USA) as described by Jimenez et al. [18]. A FlashGel DNA Marker (Lonza Inc., Rockland, ME, USA) with fragment sizes ranging from 100 bp to 4 kilo bases (kb) was used to determine the presence of the correct DNA fragments.

### 2.5. DNA Sequencing Analysis of 16S rRNA Genes in SMFC-B1

To determine the electrogenic bacterial community present in the anode of SMFC-B1, next-generation amplicon sequencing of the 465 pb 16S rRNA fragment was performed as previously described using an Illumina MiSeq protocol (Illumina, San Diego, CA, USA) [15]. An average of 58k FASTA sequences were obtained. Amplicon sequence variants (ASV) were clustered at 100% similarity [20]. The top 10 most abundant sequences were determined from each sample and homology searches were performed using the GenBank server of the National Center for Biotechnology Information (NCBI; http://blast.ncbi.nlm.nih.gov/Blast.cgi) (accessed on 24 August 2023) and the BLAST (blastn) algorithm [21].

## 3. Results

### 3.1. Electricity Generation and Electrogenic Bacteria by SMFCs

Five out of seven SMFCs generated some electricity and enriched electrogenic bacteria (Table 1 and Table 2). Start-up days for SMFCs ranged from 1 to 11 days. The average electrical generation between all seven SMFC was 155 microwatts. Only SMFC1 and SMFC-AT did not show any positive results. SMFC1 and SMFC-AT did not show any electrical production and no electrogenic bacteria. The fastest generation of electricity was obtained after 1 day by SMFC-CT. SMFC-CT showed 15 microwatts of electricity with 3.19 × 10^8^ electrogenic bacteria. SMFC-B1 showed electrical output after 1 day. However, it was half the value detected by SMFC-CT. It took SMFC-B2B 11 days to produce electricity.

The time for maximum electrical output by SMFCs ranged from 1 to 21 days. The highest electrical output was detected by SMFC-B1 with 143 microwatts after 15 days. The numbers of electrogenic bacteria for SMFC-B1 were 2.99 × 10^9^. The second-highest electrical output was shown by SMFC3 with 80 microwatts and 1.67 × 10^8^ electrogenic bacteria after 12 days.

SMFC-CT showed the longest sustainable production of electricity (Figure 2). However, after day one, the numbers for electricity and electrogenic bacteria decreased and never recovered. SMFC-CT sustained electrical production for 30 days. SMFC-B2B showed a significant electrical output after 8 days with a maximum value of 31 microwatts after 22 days. SMFC-B1 exhibited a very dynamic electricity generation. The first significant increase was detected after 8 days and then electrical output decreased with subsequent fluctuations reaching a maximum of 143 microwatts after 15 days. Finally, electrical output declined after 15 days. To determine the bacterial community in SMFC-B1, the sample was stopped after 22 days when electricity was measured to be 10 microwatts. Both SMFC-B1 and SMFC-B2B showed three cycles of fluctuations in electricity generation. It was not until after the second cycle that both devices exhibited higher electrical output.

### 3.2. 16S rRNA Analysis of SMFC-B1

Analysis of 16S rRNA genes isolated and sequenced from SMFC-B1 showed that the top ten bacteria extracted from the upper surface of the anode consisted of four Bacillota, three unknown, and three Pseudomonadota (Table 3). BLAST analysis showed that homology values ranged from 95.70 to 100%. The number one and two bacterial sequences were found to belong to an uncultured unknown bacteria. When identification was possible, four out of the top ten sequences were found to match the order Clostridiales. The other bacterial phylum detected was the Pseudomonadota with the genera *Methylocaldum* and *Magnetospirillum*.

Table 4 shows the top 10 bacteria detected under the anode. Six out of the top ten belonged to the phylum Bacillota. The rest of the sequences did not show any homology with known bacterial phyla. All six Bacillota bacteria were related to the order Clostridiales. Homology values ranged from 94.58 to 100%. When looking at the electrogenic bacteria on the anode, 80% were classified as uncultured either with some matches with known phyla, such as Bacillota or Pseudomonadota, or completely unidentified.

## 4. Discussion

Electrogenic bacteria present in SMFCs oxidized organic compounds in soil to produce electricity. The oxidation of organic substances provided electrons to the anode to produce electricity while protons migrate to the cathode through the soil. For each electron produced as an electrical current, a proton is also produced. Previous studies by our laboratory using soil samples from different locations at the BCC campus reported an SMFC with the highest electrical output of 80 microwatts and 1.67 × 10^9^ electrogenic bacteria [15]. Optimal electricity generation time was reported to be 12 days. However, in this study, SMFC3, constructed in the spring of 2023, generated similar electrical output and electrogenic bacteria. When other soils were tested during the summer of 2023, higher electrical production and electrogenic bacteria were detected. Furthermore, a longer operational time of 30 days was recorded. This was longer than any other SMFC reported by our laboratory, i.e., 23 days [15]. Of the two SMFCs started on 6/1/2023, SMFC-B1 showed 1.5 times more electrical output and electrogenic bacteria than SMFC3. A series of fluctuations in the production of electricity were generated by SMFC-B1 and it was not until after 15 days that a maximum of 143 microwatts and 2.99 × 10^9^ electrogenic bacteria were detected. Initial electrogenic activity might have relied on the oxidation of the native organic substances in the soil. But those substances were significantly depleted and it was not until other members of the bacterial community were capable of producing metabolic intermediates such as organic acids that electrogenic bacteria had additional organic substances to serve as electron generators to the anode. Soil organic carbon, mineralization rates, and bacterial community structure have been demonstrated to impact the performance of SMFCs [7,14]. Because of the closed system used in this study, depletion of natural substrates led to the eventual reduction in electricity production and electrogenic bacterial numbers. This was also shown by the fluctuations in electrical output and electrogenic bacterial numbers with some of the SMFCs indicating the possible temporary depletion of organic material. In those SMFCs, e.g., B1 and B2B, the numbers of electrogenic bacteria and electricity showed three fluctuation cycles ending with the end of electrical output and device operation.

How can we compensate for the depletion of organic substances in SMFCs? One strategy is the development of self-contained hybrid plant–soil MFCs (PSMFCs). These PSMFCs were shown to provide continuous addition of carbon substances by photosynthesis to compensate for the loss of organic matter and subsequent decay of electrical production [11]. Other studies demonstrated that electrode spacing and the addition of external organic carbon can also optimize electrical output and electrogenic bacterial numbers [13,22,23]. Substrate addition to the SMFC to compensate for the loss of organic material was observed to be better applied when electricity generation was decreasing instead of having a continuous system [13,23]. The quality of the available organic matter in soil affected the performance of SMFCs constructed from agricultural and forest soils [6]. SMFCs from agricultural soils showed 17 times more electricity than forest soils with respiration rates 10 times higher. A higher concentration of water-soluble polyphenols in forest soils compared to agricultural soils may have reduced the availability of organic matter to optimize microbial activity. Furthermore, in another study soils with high clay content and organic matter concentration supported faster and higher electrogenic activity in SMFCs [13]. High clay content provided a stronger barrier to prevent the diffusion of oxygen into the anode facilitating the development of anaerobic conditions. Soil characteristics related to electrical production by electrogenic bacteria will be further investigated by analyzing the chemical and physical composition of the soils used to generate SMFC-B1.

To understand the electrogenic bacteria composition present in SMFC-B1, 16S rRNA analysis was performed on the microbial DNA extracted from the anode. Bacterial communities in soils at BCC were shown to be predominantly comprised of the phyla Actinomycetota, Pseudomonadota, Chloroflexota, Acidobacteriota, and Planctomycetota [15]. Because the anode was buried in the SMFC, samples were taken from the biofilm on (top) and under the electrode. The 16S rRNA sequences with the highest frequencies in both locations were unknown bacteria which demonstrated the inability of the current databases to identify the microbial dark matter [24]. Microbial dark matter can be a major source of new biosynthetic pathways and enzymes with biotechnological applications in different areas. Furthermore, some of the predominant electrogenic bacteria in SMFC-B1 were related to uncultured bacteria with no cultured reference isolate available.

When bacterial identification was possible, the Bacillota were the number one bacterial phylum accounting for most of the 16S rRNA sequences in the anode. The class Clostridia was found to be responsible for all the Bacillota sequences. The members of the class Clostridia have been shown to be important contributors in SMFCs during electricity generation either by directly generating electrons transferred to the anode or by producing organic acids that were subsequently oxidized by other electrogenic bacteria within the anerobic environment of the SMFC [6,15,16,23,25]. Previous studies in our laboratory demonstrated an increase in Bacillota in the SMFC when compared to soil. Only SMFCs with a significant increase in anaerobic Bacillota generated electricity and enriched electrogenic bacteria [15]. However, in this study we developed an SMFC, i.e., B1, with the highest electrical output and electrogenic bacteria from BCC soils. In addition to the Bacillota, two other Pseudomonadota genera were detected to be part of the dominant community. They were *Methylocaldum gracile* and *Magnetospirillum* sp. These two genera were not detected in previous studies by our laboratory when different soil samples were tested [15]. *Magnetospirillum* sp. can ingest iron and proteins inside the cells interact with it to produce magnetite that is located inside membranous structures called magnetosomes. Most electrogenic bacteria such as *Clostridium* sp. and *Magnetospirillum* sp. showed the ability to reduce Fe(III). However, some can also reduce nitrate. *Magnetospirillum* sp. was shown to be predominantly present in the anodic biofilms developed by an anaerobic sludge-MFC [26]. Furthermore, when present along with *Clostridium* sp., these bacteria optimized electricity generation. SMFCs developed from Chinese soils contained a predominantly electrogenic bacterial community comprised of bacteria belonging to the class Clostridia that were capable of reducing Fe(III) [25]. The researchers of that study reported that the family Clostridiaceae represented the dominant electrogenic bacteria in soils. SMFCs constructed with German soils showed an increase in the abundance of bacteria belonging to the Bacillota phylum when electricity production was the highest [23]. In that study, they found that the electrode materials were the most important factor for sustainable electrical generation. Modified stainless steel produced optimal electrical generation compared to carbon felt. They concluded that microbial diversity and soil chemistry were not as important during the optimization of electrical production.

The other Pseudomonadota found in the anode was *M. gracile*. *M. gracile* are methanotrophic bacteria capable of methane oxidation under low levels of oxygen [27]. Other types of methanotrophic bacteria and archaea can also oxidize methane under strict anaerobic conditions using different electron acceptors [25]. Because of the anaerobic conditions detected in SMFC-B1, organic substances were decomposed by either hydrolysis, acidification, fermentation, or methanogenesis. Methanogenesis was previously shown to compete for electrons against electrogenesis in MFCs [28,29,30]. A higher concentration of archaea in SMFCS may not be beneficial to electrogenesis [29,30]. Furthermore, the final product of the anaerobic decomposition of organic substrates in soils is methane but partial decomposition during fermentation or methane oxidation by methanotrophic bacteria produced compounds such as formate and acetate that are used by either methanogens to produce methane or by electrogenic bacteria such as *Clostridium* sp. that will oxidize these compounds and transfer the electrons to the anode [27,29].

We will continue sampling additional soil locations to determine their potential to develop a vigorous and sustainable electrogenic bacterial community to optimize electrical output. Future studies in our laboratory will try to isolate the predominant electrogenic uncultured bacteria from the SMFCs. The predominant bacteria in SMFC-B1 were unculturable and their superior electrogenic ability can be optimized by trying to isolate them using selective strategies [31]. For instance, electrogenic bacteria were isolated using in situ electrodes where the anode was inserted directly into a borehole in a mine for the growth of electrogenic sulfur-reducing bacteria [32]. In SMFCs, we can exploit the naturally existing redox gradient and the fact that the anode is buried in the anaerobic part of the device while the cathode is above the soil. Enrichment broths with insoluble Fe^+3^-oxides have been also used as electron acceptors to isolate viable electrogenic bacteria [32]. Another possibility is using an ‘electroplate method’ where a diluted cell suspension is streaked on agar plates with a soluble electron donor [33]. The plate also has a transparent anode at the top as a solid-state electron acceptor. Because most electrogenic bacteria are anaerobic, conditions must be maintained during isolation procedures to avoid exposure to O_2_. A protocol for the enrichment of electrogenic bacteria was recently reported by combining in situ electrode colonization, electrochemical enrichment, biofilm detachment, liquid dilutions to extinction, and dilutions to extinction on solid media through electrode-plating [31].

## 5. Conclusions

Several SMFCs were developed from soils located at BCC. Of the seven soils used to develop SMFCs, five showed enrichment of electrogenic bacteria and subsequent production of electricity. The start-up time, i.e., enrichment period, to develop a substantial biofilm of anaerobic bacteria and electrical output ranged from 1 to 11 days. Compared to previous studies, higher electricity generation and electrogenic bacterial numbers were detected and genetically characterized from an SMFC, SMFC-B1. Furthermore, some SMFCs showed longer operational times demonstrating the sustainability of the electrogenic bacterial community and the optimization of organic compounds to be used as electron donors to generate electricity. Bacteria belonging to the Bacillota phylum were found to be the most abundant classes, order, and genera leading to the optimization of electrical output by either generating organic compounds to provide other electrogenic bacteria substrates to be oxidized or transferring electrons to the anode. However, some members of the phylum Pseudomonadota were also needed indicating the cooperation between different bacterial populations to optimize biochemical processes to promote the enrichment of electrogenic bacteria leading to the enhancement of electrical output by some of the SMFCs. Most bacteria did not have cultured representatives, indicating the need to optimize the isolation and cultivation of electrogenic bacteria to characterize their physiological capability in the laboratory with the ultimate goal of developing MFCs with defined bacterial cultures. The high diversity of soil microbial communities might provide the possibility of developing SMFCs with higher electricity production and longer operational time. There must be novel microorganisms in other soil locations around BCC not yet cultured or identified which can be used to optimize SMFC performance. Therefore, continuous sampling of additional soil locations will determine their potential to optimize electricity generation. A better understanding of the anode microbial community and soil chemistry will contribute to the ongoing optimization of electrical production by SMFCs.

## Figures and Tables

**Figure 1 biotech-12-00065-f001:**
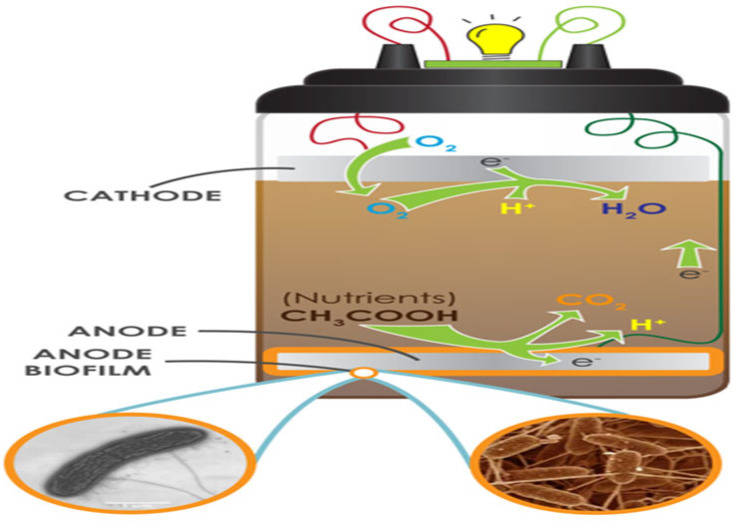
A diagram of an SMFC (Source: Adapted from [17]).

**Figure 2 biotech-12-00065-f002:**
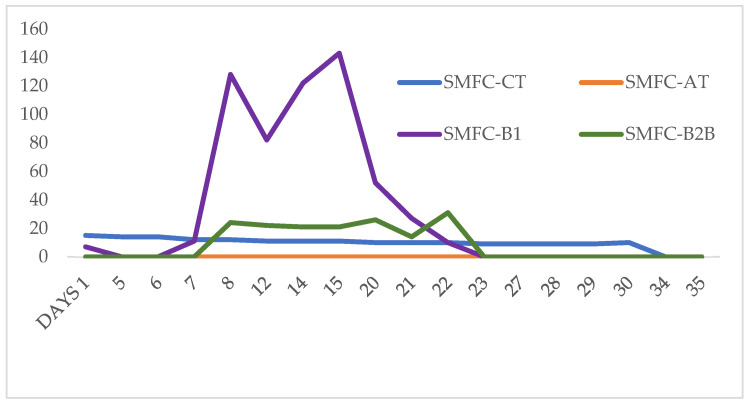
Electricity (Microwatts) generation over days by SMFCs started in June 2023.

**Table 1 biotech-12-00065-t001:** Electrical output by SMFCs.

Sample	Date	Electricity (S)	Microwatts	Electricity (H)	Microwatts
SMFC1	10 February 2023	0	0	0	0
SMFC2	10 February 2023	5	6	7	20
SMFC3	10 February 2023	3	13	12	80
SMFC-B1	1 June 2023	1	7	15	143
SMFC-B2B	1 June 2023	11	24	21	31
SMFC-CT	21 June 2023	1	15	1	15
SMFC-AT	21 June 2023	0	0	0	0

S = Start day. H = Highest day.

**Table 2 biotech-12-00065-t002:** Numbers of electrogenic bacteria in SMFCs.

Sample	Date	EBS	EB	EBH	EB
SMFC1	10 February 2023	0	0	0	0
SMFC2	10 February 2023	5	1.37 × 10^8^	7	4.33 × 10^8^
SMFC3	10 February 2023	3	2.71 × 10^8^	12	1.67 × 10^9^
SMFC-B1	1 June 2023	1	1.51 × 10^8^	15	2.99 × 10^9^
SMFC-B2B	1 June 2023	11	5.08 × 10^8^	21	6.53 × 10^8^
SMFC-CT	21 June 2023	1	3.19 × 10^8^	1	3.19 × 10^8^
SMFC-AT	21 June 2023	0	0	0	0

EB = Electrogenic bacteria. EBS = Electrogenic bacteria start day. EBH = Electrogenic bacteria highest day.

**Table 3 biotech-12-00065-t003:** Most abundant 16S rRNA sequences in SMFCB1 on anode.

Accession Number	Identification	Phylum	%Homology	Absolute Abundance N = 63,317
KC853576.1	Uncultured bacterium	U	98.83	1970
AB517723.1	Uncultured bacterium	U	93.84	1599
KX6722654.1	Uncultured Clostridia	B	98.51	829
JN540262.1	Uncultured Clostridiales	B	100	533
JN540220.1	Uncultured Clostridiales	B	97.53	416
HE804616.1	Uncultured bacterium	U	95.70	290
OQ678253.1	*Methylocaldum gracile*	P	100	258
JQ731734.1	Uncultured *Magnetospirillum*	P	100	256
MH686102.1	*Magnetospirillum*	P	98.76	250
NR_151894.1	*Anaerotaenia torta*	B	98.01	228

U = Unknown. B = Bacillota. P = Pseudomonadota.

**Table 4 biotech-12-00065-t004:** Most abundant 16S rRNA sequences in SMFCB1 under anode.

Accession Number	Identification	Phylum	%Homology	Absolute Abundance N = 52,502
AB517723.1	Uncultured bacterium	U	93.00	454
KX672654.1	Uncultured Clostridia	B	98.51	368
EU887985.1	Uncultured Clostridia	B	100	326
MH045958.1	Uncultured Clostridia	B	98.01	253
KF630866.1	Uncultured bacterium	U	100	238
NR_151894.1	*Anaerotaenia torta*	B	98.01	214
EU097334.1	Uncultured Clostridium sp.	B	99.50	192
JN540220.1	Uncultured Clostridiales	B	97.53	184
MN209875.1	Uncultured bacterium	U	94.58	178
MN209875.1	Uncultured bacterium	U	95.05	128

U = Unknown. B = Bacillota.

## Data Availability

The data presented in this study are available on request to the corresponding author.

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
