# Peer review of "Detection and Characterization of Electrogenic Bacteria from Soils"

_biotech, 2023, doi:10.3390/biotech12040065_

Round 1

Reviewer 1 Report

Comments and Suggestions for Authors

The manuscript is in line with the journal's focus areas.

Abstract

What informed the choice of sampling location (Bergen Community College or nearby areas located in the state of New Jersey)? The abstract lacks a methodological component, strengthening is needed.

Introduction

Why don't the authors hereafter use the modern taxonomy of microorganisms presented for example here - https://gtdb.ecogenomic.org/?

Material and Methods

Why do the authors cite Figure 1? How scientific is it?

It is not specified which sequencing platform (NGS) was used to obtain the results.

Results

Required to improve the quality of Figure 2.

Line 197 Error in the title of the section. Check and further in the text carefully.

The authors note, particularly in Table 4, a number of unculturable bacteria. Have they tried to cultivate them themselves? How true is this statement?

Discussion

This section is good written. However, it is possible to strengthen the relevance of the study of the mentioned dark matter by examples, if any, when scientists first discovered unculturable electrogenic bacteria at the level of genetics, and then managed to select their cultivation conditions and prove their potential. In modern microbiology, as we know, there are such examples, for example, in halophilic archaea - https://www.nature.com/articles/nmicrobiol201781.

Author Response

Reviewer 1:

Abstract

What informed the choice of sampling location (Bergen Community College or nearby areas located in the state of New Jersey)? The abstract lacks a methodological component, strengthening is needed.

Sample locations were randomly selected.  A sentence was added to the abstract to clarify the selection process.  A methodological component was added to the abstract as recommended.

Introduction

Why don't the authors hereafter use the modern taxonomy of microorganisms presented for example here - https://gtdb.ecogenomic.org/?

The names of the phylum for all bacteria were updated using the new taxonomical system.

Material and Methods

Why do the authors cite Figure 1? How scientific is it?

Figure 1 was deleted and replaced with diagram of SMFC.

It is not specified which sequencing platform (NGS) was used to obtain the results.

A sequence platform (NGS) was added to the materials and methods section.

Results

Required to improve the quality of Figure 2.

Figure 2 was improved.

Line 197 Error in the title of the section. Check and further in the text carefully.

Error corrected.

The authors note, particularly in Table 4, a number of unculturable bacteria. Have they tried to cultivate them themselves? How true is this statement?

We did not try to cultivate the unculturable bacteria.  That was beyond the scope of the work. However, future studies as mentioned in the discussion will attempt isolation and cultivation.

Discussion

This section is good written. However, it is possible to strengthen the relevance of the study of the mentioned dark matter by examples, if any, when scientists first discovered unculturable electrogenic bacteria at the level of genetics, and then managed to select their cultivation conditions and prove their potential. In modern microbiology, as we know, there are such examples, for example, in halophilic archaea - https://www.nature.com/articles/nmicrobiol201781.

Examples of previously unculturable electrogenic bacteria and selective cultivation techniques were discussed.  See lines 390-406.  References 31, 32, and 33.  As per the discussion, future studies in our laboratory will try to cultivate some of the unculturable electrogenic bacteria.

Reviewer 2 Report

Comments and Suggestions for Authors

The present manuscript “Detection and Characterization of Electrogenic Bacteria from Temperate Soils” have the author studied that an analyzed different soils around the Bergen Community College (BCC) campus / or nearby locations, located in the state of New Jersey, USA to determine the microbial community potential to generate electricity and to identify electrogenic bacteria directly by 16S rRNA analysis. These bacteria belonging to the phylum Proteobacteria ((15%) and Unidentified uncultured bacteria accounted for 35% of the predominant bacterial community in the anode of Soil microbial fuel cells (SMFCs). This work contributes to the understanding of the distribution and diversity of electrogenic bacteria in temperate soils and the optimization of electricity generation by bacteria. The reported work is encouraging and more precisely fits into the Journal. Before the manuscript deems acceptable for publication, it needs be revised very well. Some specific suggestions are listed below.

 1.Introduction is somewhat monotonous and can be improved using the recent literature.

2. Authors already published same research area articles like that https://www.jstor.org/stable/26991520.  Authors must define clearly the novelty in your paper compared to other similar works.

3. Need to more discuss future perspectives related research work.

4. Rewrite the Conclusion section, for more clearly mention the novelty of the paper.

5. The authors must be rechecking the English of the manuscript. It is fairly poor in some parts of the paper and must improve it.

Comments on the Quality of English Language

It is necessary to refine the English language.

Author Response

Reviewer 2:

The present manuscript “Detection and Characterization of Electrogenic Bacteria from Temperate Soils” have the author studied that an analyzed different soils around the Bergen Community College (BCC) campus / or nearby locations, located in the state of New Jersey, USA to determine the microbial community potential to generate electricity and to identify electrogenic bacteria directly by 16S rRNA analysis. These bacteria belonging to the phylum Proteobacteria ((15%) and Unidentified uncultured bacteria accounted for 35% of the predominant bacterial community in the anode of Soil microbial fuel cells (SMFCs). This work contributes to the understanding of the distribution and diversity of electrogenic bacteria in temperate soils and the optimization of electricity generation by bacteria. The reported work is encouraging and more precisely fits into the Journal. Before the manuscript deems acceptable for publication, it needs be revised very well. Some specific suggestions are listed below.

 1.Introduction is somewhat monotonous and can be improved using the recent literature.

Recent literature was added to the introduction.  Introduction was modified as recommended.

  1. Authors already published same research area articles like that https://www.jstor.org/stable/26991520.  Authors must define clearly the novelty in your paper compared to other similar works.

     The novelty in this study is based upon the facts that we showed new sampling sites with higher electrical output, higher numbers of electrogenic bacteria, and/or longer operational times than the previous study cited in reference 15.

  1. Need to more discuss future perspectives related research work.

     Future studies were discussed.  See. Lines 331-333. Lines 388-392. Lines 428-431.

  1. Rewrite the Conclusion section, for more clearly mention the novelty of the paper.

Conclusion was rewritten to emphasize the novelty of the work.

  1. The authors must be rechecking the English of the manuscript. It is fairly poor in some parts of the paper and must improve it.

The English grammar, spelling, and format were checked and modified using Microsoft spelling and grammar application.

Reviewer 3 Report

Comments and Suggestions for Authors

My first and primary concern lies in the novelty of this work, as I feel that the novelty issue has not been sufficiently highlighted in the current version. An important question shall be answered: does this work fill up some knowledge gaps which previous articles cannot address?

Comment 1: I suggest rewriting the topic. From the temperate latitudes of 23.5° to 66.5°, '7 soils from Bergen Community College', can it represent the whole temperate zone?

Comment 2: Eliminate multiple references. After that please check the manuscript thoroughly and eliminate all the lumps in the manuscript. This should be done by characterising each reference individually. This can be done by mentioning 1 or 2 phrases per reference to show how it is different from the others and why it deserves mentioning.

Comment 3: In the introduction, you need to connect the state of the art to your paper goals. Please follow the literature review by a clear and concise state of the art analysis.  This should clearly show the knowledge gaps identified and link them to your paper goals, by citing relevant references, e.g., Biotechnology for Biofuels, 2019, 12: 160; Environment International, 2023, 177: 108035.

Comment 4: Why the electricity generation of SMFC is different, and the basic physical and chemical properties of soil are different? What are the key influencing factors?

Comment 5: Different temperatures appear in lines 11 and 113.

Comment 6: This study was conducted at 35 °C or 37 °C, and relevant studies have shown that the higher the temperature, the higher the electrical efficiency, why not culture at normal room temperature, such as 25 °C.

Comment 7: Please supplement the device diagram of SMFCs in this study.

Comment 8: Why is not Table 1 shown as Figure 2? What is the difference between Table 1 and Figure 2?

Author Response

Reviewer 3:

My first and primary concern lies in the novelty of this work, as I feel that the novelty issue has not been sufficiently highlighted in the current version. An important question shall be answered: does this work fill up some knowledge gaps which previous articles cannot address?

Comment 1: I suggest rewriting the topic temperate. From the latitudes of 23.5° to 66.5°, '7 soils from Bergen Community College', can it represent the whole temperate zone?

The word temperate was deleted from title and text.  Any reference to temperate zones was deleted from text.

Comment 2: Eliminate multiple references. After that please check the manuscript thoroughly and eliminate all the lumps in the manuscript. This should be done by characterising each reference individually. This can be done by mentioning 1 or 2 phrases per reference to show how it is different from the others and why it deserves mentioning. 

Manuscript was checked as recommended as modified accordingly.

Comment 3: In the introduction, you need to connect the state of the art to your paper goals. Please follow the literature review by a clear and concise state of the art analysis.  This should clearly show the knowledge gaps identified and link them to your paper goals, by citing relevant references, e.g., Biotechnology for Biofuels, 2019, 12: 160; Environment International, 2023, 177: 108035.

Introduction was changed as recommended and new references on soil microbial fuel cells were added to the discussion.

Comment 4: Why the electricity generation of SMFC is different, and the basic physical and chemical properties of soil are different? What are the key influencing factors?

We did not analyze the physical and chemical properties of the soils used to generate the SMFC.  However, future studies will look into it.  See lines 331-332. The soils used were not anaerobic soils such as sediments.  They were surface soils.  To generate electricity in the system you needed anaerobic conditions as stated in the discussion Lines 345-373.  The predominant electrogenic bacteria were anaerobes member of the phylum Bacillota.

Comment 5: Different temperatures appear in lines 11 and 113.

Temperatures were harmonized through the text.

Comment 6: This study was conducted at 35 °C or 37 °C, and relevant studies have shown that the higher the temperature, the higher the electrical efficiency, why not culture at normal room temperature, such as 25 °C.

A previous work, reference 11, showed higher electrical generation with soils was detected at 35-37C.

Comment 7: Please supplement the device diagram of SMFCs in this study.

A device diagram was added to the text as Figure 1.

Comment 8: Why is not Table 1 shown as Figure 2? What is the difference between Table 1 and Figure 2?

Table 1 showed the time where all seven SMFC started producing electricity and the time where the highest value was detected.  Figure 2  showed electricity over time for samples set up in June 2023.  You can see the dynamics of electrical output.
